# Between Stress and Response: Function and Localization of Mechanosensitive Ca^2+^ Channels in Herbaceous and Perennial Plants

**DOI:** 10.3390/ijms222011043

**Published:** 2021-10-13

**Authors:** Félix P. Hartmann, Erwan Tinturier, Jean-Louis Julien, Nathalie Leblanc-Fournier

**Affiliations:** Université Clermont Auvergne, INRAE, PIAF, 63000 Clermont-Ferrand, France; erwan.tinturier@uca.fr (E.T.); J-Louis.Julien@uca.fr (J.-L.J.)

**Keywords:** biomechanics, mechanobiology, mechanosensing, thigmomorphogenesis, calcium channels, mechanosensitive channels, MCA, Piezo, OSCA

## Abstract

Over the past three decades, how plants sense and respond to mechanical stress has become a flourishing field of research. The pivotal role of mechanosensing in organogenesis and acclimation was demonstrated in various plants, and links are emerging between gene regulatory networks and physical forces exerted on tissues. However, how plant cells convert physical signals into chemical signals remains unclear. Numerous studies have focused on the role played by mechanosensitive (MS) calcium ion channels MCA, Piezo and OSCA. To complement these data, we combined data mining and visualization approaches to compare the tissue-specific expression of these genes, taking advantage of recent single-cell RNA-sequencing data obtained in the root apex and the stem of *Arabidopsis* and the *Populus* stem. These analyses raise questions about the relationships between the localization of MS channels and the localization of stress and responses. Such tissue-specific expression studies could help to elucidate the functions of MS channels. Finally, we stress the need for a better understanding of such mechanisms in trees, which are facing mechanical challenges of much higher magnitudes and over much longer time scales than herbaceous plants, and we mention practical applications of plant responsiveness to mechanical stress in agriculture and forestry.

## 1. Mechanical Stresses and Responses in Plants

### 1.1. Ubiquity of Mechanical Stress in Plants

During their development, plants are exposed to changing and fluctuating mechanical forces, both external and internal [1]. Above ground, transient mechanical loads are caused chiefly by wind [2], but also by rainfall, snow, animals and neighboring plants. Gravity acts permanently, but its magnitude increases with the plant mass and the seasonal bearing of fruits [3], and the lever arm of the weight fluctuates when the plant is swaying in the wind [4]. Under ground, the soil opposes a mechanical resistance to root penetration [5]. In addition, forces exerted above ground are transmitted to the soil, which in turn exerts reaction forces on the roots. Sessile algae and other aquatic plants are supported by buoyancy, which counteracts their weight, but they are dragged by currents [6] or tidal flows [7], and agitated by eddies.

Even in the absence of external forces, plants are mechanically stressed structures [8]. The primary source of intrinsic stress in plants is turgor pressure in cells, which is counterbalanced by stress in the cell wall [9,10]. Within a tissue, a strong stress heterogeneity is possible between neighboring cells [11], due to differences in cell size or shape [12,13,14], but also cell neighbor number [15].

Because cells are rigidly interconnected through their middle lamella, stress is transmitted from one cell to the other and builds up at the tissue scale [16]. This self-induced stress in the tissues is called “tissue stress’’ [17], “residual stress’’ [18], or “autostress” [19] (biologist-friendly definitions of mechanical terms can be found in [20]). Adjacent tissue layers can be subjected to mutual tension due to tissue shape and/or anatomical differences. In the shoot apical meristem (SAM) of tomato, the outer cell wall of the epidermal layer is about seven times thicker than the other cell walls [21]. Being stiffer, the epidermal layer bears much of the load generated by the turgor of the inner tissue and concentrates the stress. This is why the SAM has been described mechanically as a shell under pressure [22,23]. Similarly, the sunflower hypocotyl, where the outer tissue is stiffer and under tension while the inner tissue is compressed, has been described as a cylindrical pressure vessel [24,25]. In these examples, the stress pattern can be predicted based on tissue shape (Figure 1A). In the epidermis of the elongating *Arabidopsis* hypocotyl, however, Verger et al. [16] found that the longitudinal stress is higher than the transverse stress, which contradicts the cylindrical pressure vessel model. In this example, the stress pattern is not just based on tissue shape, but predominantly on the anisotropic growth of the inner tissues, which pulls the epidermis in the longitudinal direction (Figure 1B).

In woody species, additional tissue stress is generated during the maturation of the secondary cell wall [26,27]. These maturation stresses, when they are asymmetric, act as adaptive “plant muscles” in the trunk and branches to control posture and cope with mechanical disturbances [28]. In several tree species, a similar mechanical role is played by the bark, although based on a different mechanism [29].

### 1.2. Morphogenesis and Responses to Internal Mechanical Stress

Turgor pressure is the motor of cell growth, but cell growth induces water influx, which in turn dilutes osmotica in the cell and thus reduces turgor. Therefore, sustaining the osmohydraulic motor requires a constant adjustment of osmoticum concentration. This adjustment is thought to involve some sensing of the mechanical state of the cell wall (its thickness, strain, or stress) and a feedback mechanism on osmoticum influx and efflux [30].

Apart from sustaining cell turgor, responses to mechanical stress are ubiquitous in plant morphogenesis. A mechanical instability can be the trigger of a reaction cascade leading to a final shape, for instance, the formation of lobes in epidermal pavement cells [31]. Cell pressurization generates hotspots of compressive stress in the anticlinal walls, which can cause sudden and localized bendings of the wall, a phenomenon termed “buckling’’. The small displacement caused by the buckling is then amplified by a positive-feedback mechanism involving the cortical microtubules, until a lobe is formed [31]. The global effect of lobe formation in pavement cells is to keep mechanical stresses low [13]. Could buckling also initiate leaf primordia in the SAM? Many experimental and numerical results suggest instead that outgrowths at the meristem surface follow local maxima of the plant hormone auxin [32,33,34,35]. Auxin loosens the cell wall and facilitates organ emergence [36]. However, the polarity of the auxin transporters PIN is influenced by mechanical perturbations, creating a feedback loop between auxin distribution and mechanical stress [37,38]. The outgrowth of a primordium compresses the surrounding tissues. This does not usually cause buckling, but induces the expression of the homeobox gene *SHOOT MERISTEMLESS* (*STM*), which marks the boundary domain around primordia and ensures organ separation [39]. *STM* expression is quantitatively correlated to curvature in the boundary domain, which is itself a proxy for mechanical stress. The mechanotransduction pathway leading to *STM* expression has been proven to be auxin-independent [39].

The molecular actors through which mechanical stress activates *STM* are not known. In many instances, mechanics-driven morphogenesis involves a reorientation of microtubules in response to mechanical stress [22,40,41]. In the periphery of the SAM, microtubules are perpendicular to the direction of maximal strain, whereas they are parallel to the direction of maximal strain in the boundary between the primordium and the meristem [42]. This shows that while the response of microtubules to mechanical stress or strain is not completely clear, they are central actors in mechanosensing, including shape sensing [43]. Based on in vitro experiments [44] and in silico simulations [45], Hamant et al. [46] hypothesized that microtubules are sensors of tensile stress direction, or at least sensors of changes in the tensile stress direction. Microtubules guide the deposition of cellulose microfibrils on the cell wall [47], thereby locally increasing its resistance to tension and changing the pattern of stress distribution in a tissue globally. This is how a mechanical feedback loop could arise and help organs to reach their final shape in a robust way, as neatly exemplified in the sepal by Hervieux et al. [48,49].

### 1.3. Thigmomorphogenesis and Responses to External Stress

While internal mechanical stresses play a key role in driving organogenesis until a determinate target shape is reached, external stresses are inherently unpredictable and a source of dangerous perturbations. Plants face these challenges through an adaptive development relying on appropriate responses. This is especially true for perennial plants, which experience vast changes in size and mass through their life stages, and hence large variations in external mechanical stress. Adaptation occurs mainly in terms of growth, that is the addition of material, quantitatively and qualitatively. In trees subjected to transient bendings, the root biomass increases and the root architecture changes in a way that improves anchorage [50,51]. The same treatment, mimicking the effect of wind, induces a reduced height growth and an increased diameter growth in *Ulmus americana* [52] and *Prunus avium* [53], with an ovalization of the stem cross-section along the bending direction. In poplar, the inhibition of height growth is less significant, but an ovalization has also been observed [54]. This geometrical change improves the mechanical safety margin of treated trees [54]. Qualitatively, imposed bendings modulate wood differentiation. In *Abies fraseri*, the tracheids formed under bidirectional bending treatment have smaller lumens than in normal wood and higher microfibril angle [55]. Although bending-induced wood shares some similarities with compression wood, it has a distinctive anatomical structure and has been termed “flexure wood’’ by Telewski [55]. In poplar, transient unidirectional bendings cause a decrease in vessel frequency and an increase in the thickness of the secondary cell wall of fibers [56]. Unlike bidirectional bendings, unidirectional bendings allow the observation of differential effects between the stretched and compressed sides of the bent stem. For instance, vessel size is decreased in the stretched side only [56]. Thus, while radial growth depends on the absolute magnitude of strains, wood-forming tissues have the ability to distinguish between positive and negative strains. Overall, wood produced under mechanical stress is more resistant to damage and enhances the bending strength of the stem [57].

These various manifestations of growth acclimation to mechanical perturbations are collectively termed “thigmomorphogenesis” [58]. Thigmomorphogenesis tends toward better self-support and increased resistance of trees to wind [59]. Despite its high cost in terms of biomass allocation, it is a priority requirement for trees, even under severe water stress [54]. Tree acclimation to wind loads has been demonstrated not only in controlled experiments but also in a forest context [60]. Although most striking in perennials, thigmomorphogenesis has also been reported in herbaceous plants, for instance *Arabidopsis* [61,62] and *Plantago major* [63], and aquatic plants [64].

### 1.4. Localized versus Distant Responses

Mechanical actions can be very localized and initiate equally localized responses. For example, the emergence of a lateral root can be a response to a local bending [65,66]. Lateral root emergence is also an example of mechanical stress exerted from inside. The lateral root can emerge only if the epidermis yields to internal pressure. This is mediated by a complex signaling cascade, which modulates the mechanical properties of cell walls [67]. External stresses are exerted by the soil on the progressing root cap. When a poplar root encounters a physical obstacle, its growth rate sharply decreases, even before the force builds up at the contact point [68]. This early growth response is most likely induced by the mechanosensing of very small forces on the root cap. In laboratory conditions, when the root is growing in a nutrient solution and is free to move laterally, the force exerted on the obstacle increases progressively until the root buckles [68]. If the root is laterally supported, such as in a natural environment, it can reach an axial force much larger than the buckling threshold [68].

To investigate the ability of *Arabidopsis* roots to progress in a medium with varying strength, Roué et al. [69] designed an experimental setup consisting of two layers of growth medium containing Phytagel at different concentrations. The lower layer has a higher concentration and is consequently more resistant to penetration, mimicking a more compact soil layer. Upon contact with the interface between the two layers, the root either penetrates the lower layer or reorients its growth [69]. *Arabidopsis* genotypes exhibiting contrasted root cap morphologies have different penetration abilities. The *fez-2* loss-of-function mutant, with a pointed cap, penetrates less often the denser layer than the Col-0 wild type, which in turn has a lower penetration ability than the *smb-3* loss-of-function mutant, with a rectangular-shaped cap [69]. It is yet to be determined whether these unequal penetration abilities are caused solely by differences in root cap morphology, or results from possible mutation-induced alterations of the mechanical rigidity or mechanosensitivity of the root cap. Non-penetrating roots of all three genotypes undergo a rapid curving interpreted as passive buckling. In some cases, especially when the change in medium strength across the interface is highest, the buckling deviates the root apex in the horizontal direction and, later, a second curving reorients the apex to the vertical. This second curving is interpreted as an active movement resulting from the interaction between gravitropic and thigmotropic responses [69].

Responses can also occur far away from the stimuli. In manually bent tomato plants, although only the basal part of the stem directly experiences a mechanical stress, a growth response is also observed in distant primary meristems in the form of reduced height growth [70]. Since the estimated velocity of this long-distance signaling is incompatible with hormone transport [71], it is either hydraulic [72,73] or electrical [74] in nature. In *Arabidopsis*, long-distance signaling is also observed after mechanical wounding [75]. Tissues far from the wound rapidly synthesize jasmonates, which have been proven to enhance pest resistance and regulate growth [76]. Farmer et al. [77] hypothesized that the hydraulic and electrical signals induced by a wound could be mechanically coupled. After wounding, a traveling pressure wave increases locally and abruptly turgor in xylem vessels, which in turn exert strong pressure on adjacent cells. According to the so-called “squeeze cell hypothesis’’, mechanical stress in squeezed adjacent cells activates Ca^2+^ channels, causing ion fluxes, which propagate from cell to cell.

In poplars, local flame wounding on the stem alters stem elongation and stimulates, in stem apices, the expression of *JAZ5*, a gene encoding Jasmonate-ZIM domain protein [78]. On the other hand, moderate poplar stem bending does not trigger any remote response [78] although it induces a long-distance electrical signal [74]. A possible explanation could be that a moderate bending does not cause a wound, and therefore induces a weaker signal, which attenuates over a shorter distance [74]. This would also explain why tomato plants, which are short, display a remote growth response to bending [70], whereas in bent poplar the electrical signal cannot travel the longer distance between the wounding point and the stem apex.

As a conclusion, plants respond to mechanical stress in many different ways. However, responding to a stress is only possible if the stress is perceived. Mechanosensing and mechanotransduction have hence been the focus of numerous studies using the powerful tools of molecular biology. The next two sections of this review will be devoted to the description of some molecular actors of mechanosensing and how they can be related to the growth and development of roots and stems.

## 2. Mechanosensitive Channels across the Plant Kingdom

### 2.1. Diversity of Mechanosensing Mechanisms

As described in the previous section, cells from diverse tissues detect mechanical signals through potentially diverse mechanisms, triggering different biological responses. These different types of mechanical stresses (e.g., bending, dryness or hyperosmotic shock) cause stretching of the plasma membrane, displacement of the plasma membrane relative to the cell wall (e.g., cell-wall loosening during cell expansion, hypo-osmotic shock), changes in the orientation of cortical microtubules [22] and could also affect the nucleus through interactions between cellular compartments via the cytoskeleton [79]. The perception of a mechanical stress by a cell is a rapid process requiring a quick conversion of a mechanical signal into a biological signal. Major progress has been made in recent years in identifying the molecular players involved in mechanosensing. Two major types of candidates emerge from these studies: (i) receptor-like kinases from *Catharanthus roseus* RLK1-like subfamily (CrRLK1) identified in approaches aimed at studying the maintenance of cell wall integrity (see review by Bacete and Hamman [80] and references therein), and (ii) mechanosensitive (MS) ion channels (see review by Frachisse et al. [81] and references therein).

Among the CrRLK1 family, FERONIA is a player at the interface of many responses involving the perception of mechanical stresses. Thus, besides its well-known role in cell growth and hormone signaling [82], the *fer* loss-of-function mutant has impaired ion signaling, reduced expression of mechanoresponsive genes and altered root growth responses to mechanically challenging environments when exposed to local touch or bending stimulation [83]. Furthermore, FERONIA also appears to play a role in developmental responses involving the perception of mechanical stimuli such as root nutation [84], root gravitropism [85], roots encountering an impenetrable cover glass barrier [83]. This transmembrane protein has an extracellular domain containing two MALECTIN-like domains that may be involved in binding to cell wall components and an intracellular kinase domain that may trigger phosphorylation-dependent signaling pathways during mechanotransduction. Interestingly, a recent work shows the involvement of several phosphoproteins—MAP kinase kinase (MKK) and a novel protein named TREPH1—in the process of the most-studied thigmomorphogenetic response in *Arabidopsis*, i.e., the emergence of the flowering stem [86].

The other important actors are the MS channels, which are inserted in the membrane and function as ion transporters across the membrane when under mechanical stress. They are prime candidates for converting a mechanical signal into a chemical signal. Described in various reviews [81,87,88], several types of MS channels have been identified in plants, including the anion-permeable channel MSL and several types of calcium-permeable channels families such as MCA, Piezo and OSCA.

Another way to understand the early stages of mechanosensing is to follow the dynamics of intracellular calcium concentrations. Indeed, calcium cations are typically involved as quick secondary messengers in numerous plant responses [89]. A rapid increase in concentration of cytosolic Ca^2+^ ions occurs in response to mechanical stimuli such as touch, wind or osmotic variations in plants [90,91,92,93], triggering a specific local response, but also possibly causing distant responses by propagating electrical signals [94,95]. The diversity of calcium signatures depending on the nature of the mechanical stimuli is very well described in the study carried out by Monshausen et al. [67] on the *Arabidopsis* root. In this study, the authors showed that touch, bending or root growth against an impenetrable barrier trigger a rapid (<30 s) increase in intracellular calcium. However, the calcium signatures are different in duration and in the peak intensity, suggesting the potential involvement of different mechanosensing mechanisms or the importance of a sensitivity threshold. It is also important to note that, in the case of bending stimulus, the calcium increase is visible mainly in the tensile tissues of the root, suggesting potentially different sensing mechanisms between compressive and tensile strains. Moreover, the calcium response is biphasic, involving two peaks of different intensities. The channels involved in this calcium response have not been identified to date. However, FERONIA seems to be involved in this response since *fer* loss-of-function mutants subjected to such mechanical stimulation do not show the second calcium peak, characteristic of this biphasic response. However, the calcium-permeable channels responsible for the first calcium peak and the role of FERONIA in triggering the second calcium peak remain to be identified. Shih et al. [83] report that under their conditions, the loss-of-function mutant *mca1* and the quintuple mutant *msl4 msl5 msl6 msl9 msl10* are not impaired in this response.

Given the predominant position of the calcium-permeable channels in the literature on plant mechanotransduction, we will focus in the rest of this section on this mechanosensing mechanism. It should also be noted that the genes involved in the calcium signaling pathway such as calmodulins were among the first genes discovered to be regulated at the transcriptional level by mechanical stimuli [96].

### 2.2. Description and Functioning of Mechanosensitive Ca^2+^ Channels

#### 2.2.1. The Mid1-Complementing Activity (MCA) Family

The MCA family was first described in *Arabidopsis thaliana* [97]. *AtMCA1* and *AtMCA2* are paralogs, with high sequence homology (the relationships between gene names and accession numbers are given in
Appendix A).

The MCA channel was thought to be organized as a homotetramer with a small transmembrane region forming the pore (with only one transmembrane domain per monomer) and a larger cytoplasmic region of motifs (an EF hand-like motif, a coiled-coil motif and a cysteine-rich region homologous to the PLAC8 or DUF614 motif near the carboxyl terminus) regulating the pore [98,99,100]. While the PLAC8 domain is observed in eukaryotic proteins, its combination with the transmembrane domain and the EF hand-like motif is specific to streptophytes, and absent in chlorophytes (Figure 2A). Therefore, in contrast to other MS channels (Figure 2), MCA appears to be specific to land plants [101]. This may relate to the specific selective pressure faced by land plants, which can not rely on water buoyancy to support themselves and hence developed posture control mechanisms to maintain an erect habit [4]. However, it is not known whether *MCA* genes are directly involved in posture control.

Even if a fine analysis of the electrophysiological properties of this MS channel is lacking, *MCA1*-expressing *Xenopus laevis* oocytes enhances ionic currents following mechanical stimuli such as hypo-osmotic shock or in response to negative pressure in the patch pipette [97,102]. In vivo, AtMCA1 appears to be involved in Ca^2+^ uptake in *Arabidopsis thaliana* roots [103]. It is also thought to be crucial in the process of root penetration in a hard agar medium [97]. Indeed, using a two-phase agar method, Nakagawa et al. [97] demonstrated that the primary roots of *mca1*-null plants fail to penetrate a harder agar medium from a softer one, whereas *mca2*-null seedlings are able to grow like wild-type (WT) plants depending on the hardness of agar medium [103].

Hypergravity has the effect of inhibiting the growth of the hypocotyl of *Arabidopsis thaliana*. This effect is reduced in an *mca*-null mutant compared to WT, and is enhanced in an *MCA*-overexpressing mutant [104]. These results also suggest the involvement of MCA in gravity perception.

#### 2.2.2. The Piezo Family

First identified in vertebrates, the mechanically-activated Piezo family (Figure 2B) is conserved with homologs in prokaryotes, plants, and invertebrates [105].

In vertebrates, Piezo 1 and 2 are large structures of approximately 2500 amino-acids including many transmembrane regions. Cryo-electron microscopy (cryo-EM) studies have shown that Piezo1 possesses a tree-bladed, homotrimeric structure comprising a central ion-conducting pore module [106,107,108,109]. Each peripheral blade has 38 transmembrane regions, with the peripheral blade-like structures on the extracellular side and three beams on the intracellular side. With its distal blades and the central pore, the Piezo1 channel could have a lever-like mechanotransduction mechanism that could explain how it converts mechanical forces into a cation influx (see the illustrated model proposed in [110]).

In vertebrates, Piezo 1 and 2 are mechanotransducers able to convert a mechanical force into an increase in cytosolic Ca^2+^, enhancing various Ca^2+^-dependent downstream signaling pathways. Piezo1 can be activated by a large variety of mechanical stimulation such as poking, stretching or any local membrane tension, while Piezo2 seems to be strongly activated only by poking. In *Arabidopsis*, Mousavi et al. [111] studied the electrophysiological properties of the unique Piezo protein via a chimeric PZO1 construct including the C-terminal part of AtPiezo1 (containing the putative pore region) and the N-terminal part of the mouse mPiezo1 to allow the anchoring of the protein in the plasma membrane. They showed that this chimeric PZO1 protein is functional and acts as an MS channel when expressed in naive mammalian cells.

Piezo ion channels play key roles in mechanobiological processes in animals, for example, in vascular development or touch sensing [112]. In plants, their role is still largely unknown, with only four studies to date. In *Arabidopsis thaliana*, AtPiezo is a key element of the immune response to the cucumber mosaic virus and the turnip mosaic virus [113]. In addition, several recent works have made major advances on the potential role of AtPiezo in *Arabidopsis* root growth in responses to substrate impedance [111,114]. When *atpiezo* null-mutant plants were grown in vitro in vertical plates leading to root growth on the surface of the medium, no difference was observed from WT plants [114]. However, when the plants were grown in a horizontally placed plate with 0.8% agar, the root ability to penetrate the medium decreased by about 20% compared to WT. Similar results were obtained by Mousavi et al. [111], who showed that when seedling roots grew within the Murashige and Skoog medium, the length of the mutant roots were shorter compared to WT. Both studies suggest a direct role of AtPiezo in mechanosensing in roots. Interestingly, as detected for Atmca1 [103], in these two studies, reporter gene expression of Piezo::GUS in transgenic *Arabidopsis thaliana* plants showed that *AtPiezo* is mainly expressed in the root cap in a cross-section of the root tip.

However, in moss (*Physcomitrium patens*), PpPiezo1 and 2 are localized in the tonoplast and are thought to be involved in the regulation of vacuole shape in apical caulonemal cells [115]. These results raise the question of the role of the vacuole in mechanosensing. A study linking cellular elongation, vacuole expansion and the role of FERONIA has been carried out by Dünser et al. [116]. Using epidermal atrichoblast root cells and by combining different fluorescent dyes in cellular compartments, they showed that the cytosol shows relatively minor volume expansion during epidermal elongation compared to the relative increase in vacuolar size. By using different approaches, they investigated the possibility that apoplast acidification/cell wall loosening is sensed and provides a feedback control for vacuolar morphogenesis. Again, *fer* loss-of-function mutant vacuoles were markedly less affected by extracellular constraints of the substrate suggesting a role of a FER-dependent signaling in regulating the intracellular expansion of the vacuole. It would be interesting to analyze if Piezo could be involved in these mechanisms.

#### 2.2.3. The OSCA Family, Hyperosmolarity-Gated Calcium-Permeable Channels

OSCA is the largest family of mechanically-activated ion channels identified and conserved across eukaryotes, including fungi, animals (homologous TMEM63 channels) and plants (Figure 2C) [117,118]. In *Arabidopsis thaliana* and *Populus trichocarpa*, OSCA is composed of 15 and 17 genes, respectively, separated in four clades (Figure 3). Murthy et al. [118] tested the mechanosensitivity of channels across clades by selecting one gene from each clade in *Arabidopsis thaliana*. Electrophysiological results suggest that the channels encoded by the three genes *AtOSCA1.8*, *AtOSCA2.3*, and *AtOSCA3.1* are mechanosensitive, a trait that would not be found for the AtOSCA4.1 channel (Murthy et al., 2018). Clade 4 shares the least homology with clade 1 (Figure 3). AtOSCA4.1 may have a distinct function.

Cryo-EM studies revealed a molecular structure formed by a homodimer with two pores [119,120,121]. Work on AtOSCA1.1 and AtOSCA1.2 suggests that the resulting pores are non-selectively permeable to cations [118]. Each subunit is composed of 11 transmembrane domains, as well as a soluble cytosolic domain with an RNA recognition motif. The cytosolic domain has a distinct structural feature in the form of extended intracellular helical arms parallel to the plasma membrane. These arms are well positioned to detect a change in the lateral tension of the inner lipid bilayer sheet caused by a change in turgor pressure [122]. Membrane topology appears to be conserved across eukaryotes [118].

The physiological functions of OSCA channels remain poorly understood. Until now, several studies have implicated the *OSCA* genes in osmotic responses. However, a hypo- or hyper-osmotic shock results in a stretching or a relaxation of the plasma membrane, and can be considered as a mechanical stimulation. AtOSCA1.1 mediates osmotic signaling in guard cells and roots cells, and attenuates water transpiration and root growth in response to osmotic stress [123]. AtOSCA1.3 controls plant stomatal immunity [124]. In maize, ZmOSCA2.4 has been shown to be involved in drought tolerance in transgenic *Arabidopsis* [125]. Recently, an RNA-seq study by Procko et al. [126] suggested an involvement of the MS channels MSL and OSCA in triggering the touch-sensitive hair of the Venus flytrap (*Dionaea muscipula*). It would be interesting to see whether OSCA channels have a role in other mechanical processes.

The diversity of calcium-permeable MS channels present in plants could explain what makes a mechanically-induced [Ca^2+^]_cyt_ signaling pattern specific (Ca^2+^ footsprint), leading ultimately to a specific, graded physiological response. As very well explained in a recent review by Frachisse et al. [81], the detailed characterization of the electrophysiological properties of these different channels shows that they differ in their permeation and gating properties (activation threshold, conductance, inactivation). They can therefore trigger distinct signaling pathways and potentially be activated differentially, depending on the magnitude of the mechanical stress.

## 3. Mechanosensitive Channels in Tissues and Organs

The functions of the different families of MS channels are far from being fully understood. Several questions remain to be answered regarding their role during either development or acclimation to external mechanical stress. Are there relationships between the localization of MS channels and the localization of stress and responses? Is the number of different types of these channels in a membrane responsible for a difference in the sensitivity of cells to membrane stretch? Within a multigene family, is there redundancy between proteins or are they specifically localized in tissues? Do these different channels interact in the same signalization pathways? In relation to these questions, the most-studied family of MS channels is the anion-preferring MSL family [88]. Within this MSL family, consisting of 10 members in *Arabidopsis*, some isoforms localize in membranes of different cell compartments (plasma membrane, tonoplast, mitochondrial membrane) and are expressed in a tissue-specific manner. Concerning calcium-permeable channels, MCA and OSCA channels are located in the plasma membrane [97,123], whereas Piezo1 is located in the tonoplast [115]. However, information is incomplete regarding the location of their expression.

Studying the regulation and the localization of the expression of these genes could help to design new hypotheses on their function. In this section, through data mining and visualization, we compare the tissue-specific expression of these three families of MS channels in plants, focusing on root and stem tissues. In *Arabidopsis*, recent studies using RNA-seq approaches carried out on single cells have revealed the expression of thousands of genes in different root tissues [127] or in tissues of the floral stem [128]. We also studied the expression profiles of members of these families in poplar using RNA-seq data obtained by Sundell et al. [129] from laser dissection of stems comprising developing phloem and secondary xylem.

### 3.1. Mechanosensitive Channel Expression in the Root Tip

As root cap cells are the first to encounter obstacles, this tissue is a potential mechanosensing site. Therefore, the molecular mechanisms underlying root-obstacle avoidance has been the subject of much research in recent years. To mimic the progression of the root in the face of mechanical stresses related to soil impedance, studies were conducted using several in vitro systems: artificial root barrier systems or experimental setups consisting of two layers of growth medium containing Phytagel at different concentrations to modify the physical strength of the medium [69]. The typical responses of roots encountering an obstacle are a reduction of root growth and a resulting “step-like’’ growth pattern, with only the root tip remaining in contact with the barrier surface. By RNA-seq approaches performed on whole roots of *Arabidopsis* seedlings collected either 6 or 30 h after barrier contact, combined with reporter line and mutant studies, Jacobsen et al. [130] showed significant changes in reactive oxygen species (ROS), ethylene and auxin signaling pathways, confirming several previous studies. In particular, they demonstrated that when ethylene signaling is perturbed in ethylene-insensitive mutants, the root growth is less inhibited than in WT after barrier contact. In the same way, auxin transport effects were observed through altered bending responses, which is consistent with other studies using PIN-mutants, pharmacological assays and image analysis [131].

Meanwhile, in the last fifteen years, several studies have evidenced the role of MS channels in root responses to mechanical impedance stress. However, tissue-specific mecanosensing and the role of the different MS channels are far from being well-understood. So far, no link between the activation of auxin, ROS and ethylene signaling pathways and the activation of MS channels have been demonstrated during the root response to substrate impedance. Furthermore, in vivo root growth dynamics and the root trajectory following the contact of the root apex have not been finely analyzed in *mca*-mutants or *piezo*-mutant of *Arabidopsis thaliana*. Using different root-cap development mutants, this type of approach has shown that the shape of the root cap and the growth rate of the root are parameters that strongly influence the mechanical resistance of the root to buckling or the magnitude of the force applied by the root on the substrate [69]. If such anatomical or growth defects were found in these mutants, these parameters could influence the ability of *Arabidopsis* roots to progress in a medium with varying strength. In addition, the growth arrest observed following contact of the root with the substrate suggests a change in cell elongation, at a distance from the contact point, in the elongation zone. How is the signal transmitted? Are the innermost cells deformed and respond to this mechanical stress? Could the *OSCA* genes play a role in this response?

To study the location of the expression of these different genes in different cell types of the root, we compared their tissue-specific expression using novel visualizations of data obtained by Ryu et al. [127] with the single-cell RNA-seq method.

When using a common scale to compare the expression levels of all tested genes (Figure A1), it appears clearly that *AtOSCA2.4* and *AtOSCA3.1* reach the highest level of expression in the tissues composing the root. While *AtOSCA2.4* is most highly expressed in the outermost tissues (epidermis, cortex, lateral cap and columella), *AtOSCA3.1* expression is high in the phloem and cortex and at a lower level in the columella and epidermis. The genes *AtMCA2*, *AtOSCA1.4*, *AtOSCA1.5*, *AtOSCA1.6*, *AtOSCA1.7*, *AtOSCA2.5*, *AtOSCA4.1* and *AtPiezo* show a 10-fold lower expression level than the other genes. However, when the expression levels of these genes are plotted on a scale specific to each gene (Figure 4), very distinct expression profiles emerge. For example, *AtOSCA2.5* appears to be expressed in cap tissues, *AtOSCA1.5* in the pericycle, *AtOSCA1.7* in the phloem, *AtMCA2* in the endodermis and *AtOSCA1.4* in epidermal cells developing a root hair. Although low, the tissues with the highest amount of *AtPiezo* transcripts are the cortex, lateral cap, and xylem. These data are in agreement with the histochemical approaches using the GUS reporter gene highlighted by Mousavi et al. [111], although this transcriptomic approach, which allows a stronger differentiation of expression levels, shows that *AtPiezo* expression in the columella is relatively low. Similarly, the tissue-specific expression of *AtMCA1* in the columella and xylem was revealed by Yamanaka et al. [103]. Regarding the vascular tissues of the stem, apart from *AtOSCA2.1*, the genes of the *OSCA* family show a higher expression in the phloem in contrast to *AtMCA1* and *AtPiezo*. Concerning the *OSCA* gene family, it is interesting to observe that genes that are very close in sequence and probably originate from the same duplication event (see the phylogenetic tree Figure 3) can present very different expression profiles (e.g., *AtOSCA1.1* and *AtOSCA1.2*, or *AtOSCA2.1* and *AtOSCA2.2*). These results suggest that these proteins may be involved in different physiological functions. The expression of many *OSCA* genes in the lateral cap or columella (*AtOSCA1.1*, *AtOSCA1.3*, *AtOSCA2.3*, *AtOSCA2.5*) could encourage studies on their role in root penetration or in the perception of water stress at the root level. In the context of mechanosensing, it would also be interesting to study *AtOSCA2.4* and *AtOSCA1.4*, which show, respectively, strong and specific expression in rapidly-expanding epidermal cells. In contrast, the expression level of all these genes encoding MS channels is very low in the meristematic zone.

### 3.2. Mechanosensitive Channels Expression in the Stem

Like in the root, cells in the plant stem are sensitive to external mechanical stress during primary or secondary growth. Numerous studies showed that repeated bending of the stem triggers an increase in stem diameter growth and anatomical changes in internal stem tissues, affecting especially the development of secondary tissues [55,56,61]. However, studying the spatio-temporal responses in vivo in the tissues of a secondary-growing stem is complex and molecular studies seeking to reveal the mechanisms underlying this acclimation response to external mechanical stimuli are underdeveloped. In poplar, Pomiès et al. [132] showed that the stem transcriptome is massively remodeled in response to a single transient-bending stimulus. A total of 2663 genes, representing 6% of the genome, were found to be regulated after bending, with a large proportion (75%) of these genes being regulated within the first 2 h after bending. Short-term transcriptomic responses entailed a rapid stimulation of plant defense and abiotic stress signaling pathways, including ethylene and jasmonic acid signaling [132]. Late transcriptomic responses affected genes involved in cell wall organization and/or wood development. By studying more specifically, the stretched and compressed zones of the bent stem, Roignant et al. [56] demonstrated that some genes (e.g., *MYB69* transcription factor, microtubule-associated protein encoding gene *MAP70*) were regulated in the same way in both zones, while *FLA* genes, encoding fasciclin-like arabinogalactan proteins known to accumulate during tension wood formation, were specifically expressed in the stretched zone. These data suggest that cells sense differentially stretch and compression induced by stem bending. To date, the role of MS channels in this type of response has not been studied. However, the growth and anatomical differences that have been established show that this sensitivity to bending modifies the cambial activity within the stem. Are all living cells in the stem mechanosensitive or is the sensing localized and secondary signals are transmitted remotely to the cambium?

As in the case of the root, we took advantage of recent RNA-seq data on single cells or isolated tissues to compare the expression of the *MCA*, *Piezo* and *OSCA* genes. As shown in Figure 5, *AtOSCA4.1*, *AtOSCA2.5* and *AtMCA2* are very weakly expressed in the *Arabidopsis* stem. Within the *OSCA* family, no clade-specific expression pattern emerges from these data. Whatever the tissue analyzed, the expression of some *AtOSCA* genes is higher than *Piezo* or *AtMCA1*. In particular, *AtOSCA1.3* and *AtOSCA3.1* are highly expressed. While *AtPiezo* shows a homogeneous expression profile in the different stem tissues, *AtOSCA 3.1*, *1.3* and *2.2* have particular expression patterns with a very high level in the endodermis and the differentiating vessel elements. Interestingly, *AtMCA1* is expressed more strongly in the differentiating vessel elements than in the endodermis, a tissue involved in graviperception, while recent studies showed a role of this channel in hypergravity [104].

In order to investigate whether these tissue-specific expression patterns are found in other plant species, we also performed data mining and visualization on expression data obtained by RNA-seq after cryodissection of stem sections at the level of phloem and differentiating xylem tissues in *Populus tremula* [129]. As in *Arabidopsis*, *PtPiezo1* shows a homogeneous expression pattern in the different tissues (Figure 6). No data were found for the *PtPiezo2* and *PtOSCA2.1* genes, and the level of *PtOSCA1.4* and *1.5* expression is very low compared to the other genes. Among the genes encoding MS channels, *PtMCA1* is the most strongly expressed in the regions of expanding xylem and secondary wall development, whereas *PtMCA2* is very weakly expressed in these same regions. As in *Arabidopsis*, in the *OSCA* multigene family, the *PtOSCA3.1* isoform has the highest level of expression in the stem, but with a lower level in xylem cells during secondary wall differentiation. In clade 2, the genes are homogeneously expressed in all tissues except *PtaOSCA2.5*, for which the expression level is very low during the secondary wall differentiation phase. Finally, heterogeneous expression profiles are observed within clade 1 of the *OSCA* family: isoforms *1.1* and *1.2* show a peak of expression in the secondary wall formation zone while isoforms *1.3* and *1.7* in the phloem and isoforms *1.8* and *1.7* are more strongly expressed in the expanding xylem. Similarly, a peak of expression is detected for *PtOSCA1.9* in mature xylem cells. Note that no cambium-specific expression profile is observed for any of these genes. This study reveals more disparate expression levels between genes in the xylem expansion zone in poplar. It is interesting to note that in *Arabidopsis*, some genes also have a very specific expression profile in the differentiating xylem vessels.

## 4. Conclusions

### 4.1. Unraveling the Functions of Mechanosensitive Channels

Although major advances have been made in recent years demonstrating the essential role of mechanical stresses during a plant’s life cycle, how cells perceive these signals and which signaling pathways trigger the physiological or developmental responses are questions that still require exciting research. While the dwarfing phenotype of *fer* loss-of-function mutants made it possible to establish the link between these proteins, cell growth and/or maintenance of cell wall integrity, no particular phenotype has been reported yet, for example, in the *piezo* loss-of-function mutants under normal growth conditions. More careful investigation, possibly under stressful conditions, could reveal the altered phenotype at either the macroscopic or molecular level. In this respect, the progress made in transcriptomic studies within a cell type opens new perspectives. Indeed, knowing in which tissue an MS channel gene is specifically expressed could allow a more specific targeting of the analysis of this tissue in KO-mutants to detect phenotypes. A further option could even be to modify the expression of the MS gene specifically in this tissue.

The creation of plants simultaneously mutated for several of these channels in the same tissue should also be tested to see if these proteins have redundant functions. This type of approach would be particularly interesting in the case of the multigenic *OSCA* family comprising a large number of genes. However, the immense number of possible combinations makes a systematic exploration of multiple mutants extremely laborious (for instance, there are 6188 possible quintuple KO-mutants with the 17 *OSCA* genes of *Populus trichocarpa*). Correlations in the tissue-specific expression of MS channel genes could be used as criteria to narrow down the candidates and focus on the most promising combinations.

Finally, the respective importance of these channels during mechanosensing would require the ability to correlate their activation with the magnitude of membrane tension. In this context, pioneer studies using fluorescent nano-probe reporters of membrane tension are very promising [133].

### 4.2. Transcriptional Regulation of Mechanosensitive Channel Genes following a Mechanical Stress

In addition to the tissue-specific expressions shown here, the transcriptional regulation of the MS channel genes has been poorly studied. Some results suggest that mechanical stress stimulates the transcription of these genes. For example, Mousavi et al. [111] showed that *Arabidopsis thaliana* seedlings grown inside the Murashige and Skoog medium show a threefold higher expression of *AtPiezo* in the upper root and root cap than seedlings grown on the top of this medium. Zhang et al. [113] observed that the inoculation of a virus led to a significant increase in the expression of *AtPiezo* in petioles. In the carnivorous plant Venus flytrap, a transcriptomic study revealed that two *MSL* genes and an *OSCA* gene are upregulated in the touch-sensitive trigger hair following touch [126]. *Arabidopsis thaliana* seedlings under hypergravity conditions show upregulation of *AtMCA1* and *AtMCA2* in the hypocotyl compared to seedlings under the 1g condition [104]. Since those mechanics-related gene expression patterns can suggest a gene function in the mecanosensing pathways, it could be interesting to reproduce our data-mining approach on previous transcriptome data obtained after a mechanical stress such as those performed after a transient stem/shoot bending [132,134], after a load has been placed on a tree stem [135] or after touching rosette leaves [136]. The expression dynamics of MS channel genes would probably give precious insight into their role in the physiological response, in short and long term. This kind of approach, combining mining and visualization of large molecular datasets with mechanical analyses at the tissue level, is a good illustration of the flourishing field of quantitative plant biology [137].

If the transcriptional regulation of MS channel genes after mechanical stress is to be confirmed, what are the implications for the amount of these proteins in the membranes? Is the transcriptional regulation just a way to maintain a constant concentration of MS channels, or could it modulate the mechanosensitivity of stimulated tissues?

### 4.3. Practical Applications of Thigmomorphogenesis

The ability of plants to respond to mechanical stress is more than a theoretical curiosity. For centuries, Japanese farmers have taken advantage of this phenomenon in a practice called “mugifumi”, which consists in treading young wheat or barley plants to reduce the risk of lodging and increase yield [138]. This practical application of thigmomorphogenesis *avant la lettre* is bound to generalize and diversify as environmentally-friendly substitutes for current chemical treatments. For instance, mechanostimulating garden roses under greenhouse conditions increases branching and compactness [139], and could become an alternative to hormonal treatments in horticulture. In Poaceae, exposure to wind and imposed flexure have an impact on agronomic traits [140]. Furthermore, as several studies showed that various forms of mechanical stimuli enhanced plants’ basal defense responses (see [141] for a review), mechanostimulation could be a potential priming method for sustainable agriculture, alternative to chemical-based priming. Developing efficient applications requires new phenotyping techniques for measuring mechanical traits such as the resistance to lodging [142], but also a deeper understanding of the actors involved in the mechanosensing pathways.

The relevance of thigmomorphogenesis to tree growth in non-controlled conditions [60] also opens new avenues for forest management, especially given the current increase in frequency of extreme windstorms [143]. At the macroscopic level, the functional biomechanical traits quantifying the strength and safety of trees against wind and self-buckling are well-established [144]. Trees acclimate to their mechanical environment by constantly adjusting these traits thanks to mechanosensing. However, the molecular actors involved in this feedback loop are largely unknown. Indeed, while many molecular studies have focused on the outermost tissues of organs (root cap, epidermis of the shoot apical meristem), few studies targeted the xylem, phloem or endodermis of stems, even though some of the MS channels genes are highly expressed in these tissues. The difficulty of performing in vivo spatio-temporal analyses in secondary tissues partly explains this lack of data. A future solution could be experimental setups for growing cambium in vitro, mimicking the mechanical and hormonal environment of the cambium in trees. In any case, perennial plants are the systems in which investigating the localization and functions of MS channels is the most ecologically relevant.

## Figures and Tables

**Figure 1 ijms-22-11043-f001:**
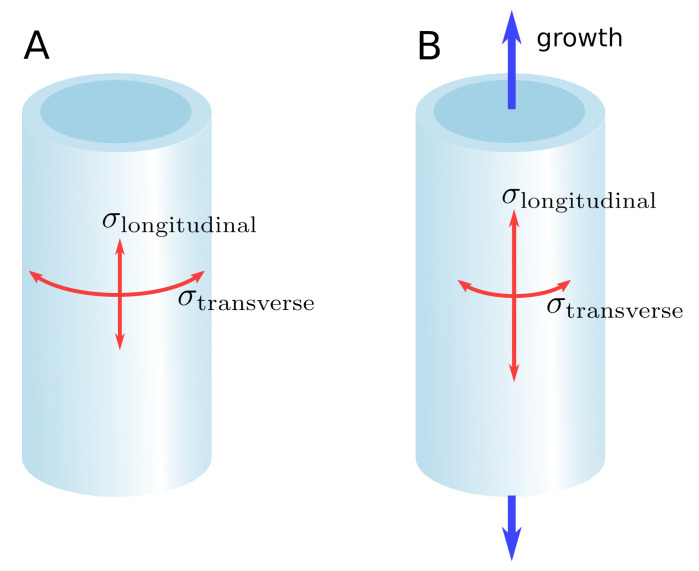
Shape-derived versus growth-derived stress pattern. (**A**) In the pressure vessel model, the stress pattern in the epidermis is determined by the shape of the organ. For a cylindrical organ, the transverse stress is higher than the longitudinal stress. (**B**) Differential growth between the inner tissue and the epidermis can override the stress pattern predicted by the pressure vessel model, leading to a longitudinal stress higher than the transverse stress.

**Figure 2 ijms-22-11043-f002:**
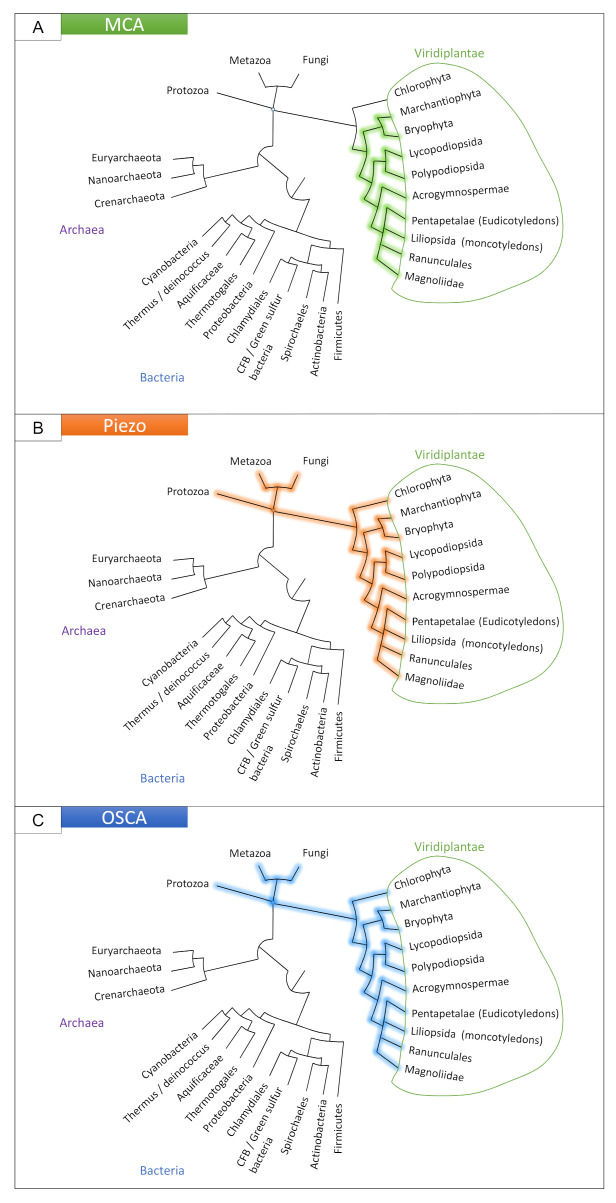
Presence of the *MCA* (**A**), *Piezo* (**B**) and *OSCA* (**C**) families in the tree of life. See Appendix B for details.

**Figure 3 ijms-22-11043-f003:**
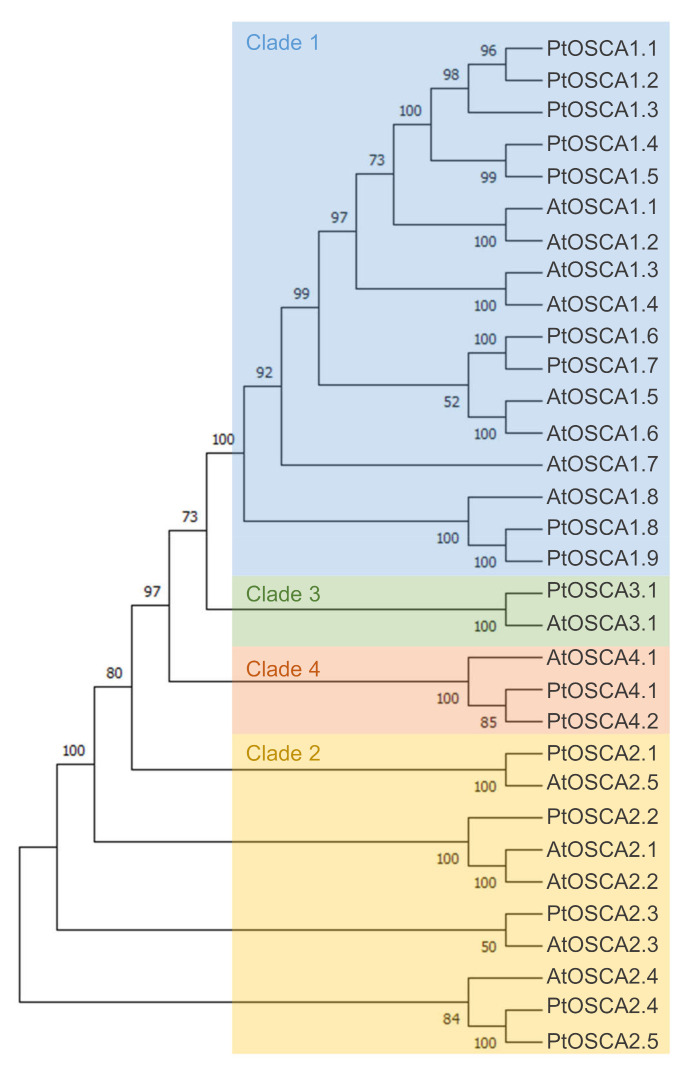
Maximum-likelihood phylogenetic tree of OSCA of *Arabidopsis thaliana* and *Populus trichocarpa*. The percentage of trees in which the associated taxa clustered together is shown next to the branches. Initial trees for the heuristic search were obtained automatically by applying Neighbor-Join and BioNJ algorithms to a matrix of pairwise distances estimated using the JTT model, and then selecting the topology with a superior log-likelihood value. The tree is drawn to scale, with branch lengths measured in number of substitutions per site. This analysis involved 22 amino acid sequences. All positions with less than 50% site coverage were eliminated, i.e., fewer than 50% alignment gaps, missing data, and ambiguous bases were allowed at any position (partial deletion option). There were a total of 770 positions in the final dataset. Evolutionary analyses were conducted in MEGA X. The four clades are colored: blue (Clade 1), yellow (Clade 2), green (Clade 3), and orange (Clade 4).

**Figure 4 ijms-22-11043-f004:**
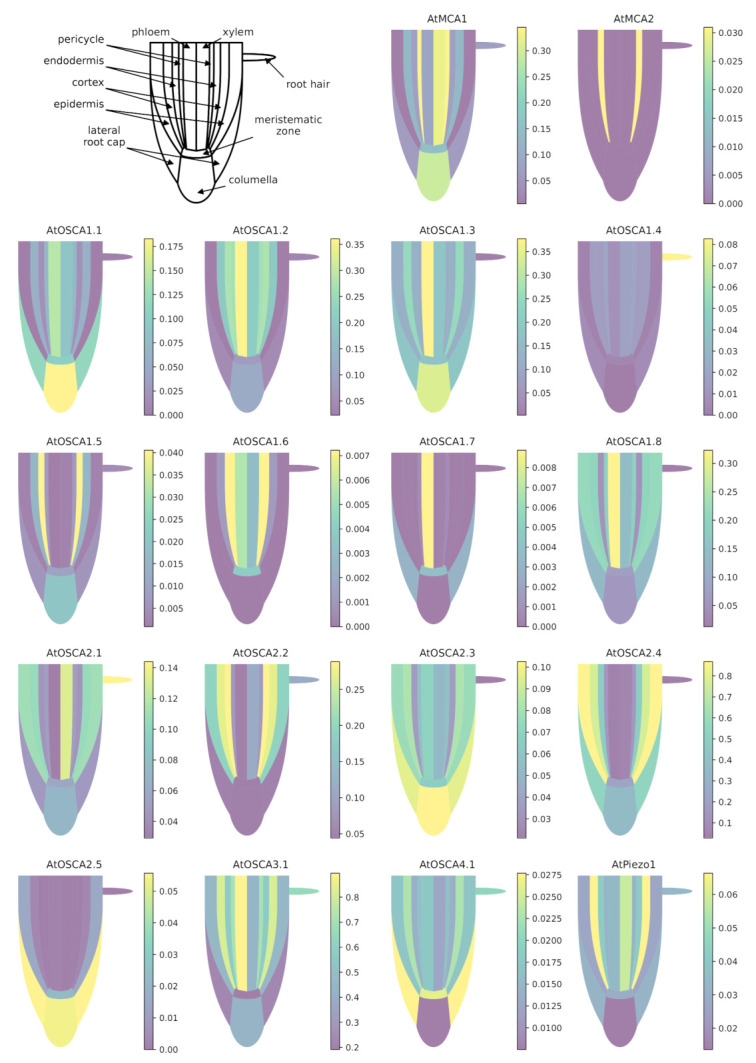
Expression patterns of *MCA*, *Piezo* and *OSCA* genes in different tissues of the *Arabidopsis* root. Data for each gene were retrieved from the BAR ePlantArabidospsis website (https://bar.utoronto.ca/eplant, accessed on 23 February 2021). To facilitate the graphical representation of expression levels in tissues, only expression values from sub-clusters containing differentiated (not differentiating) cells were used. In the stele, only the expression data from xylem- and phloem-cell types are shown. In the meristematic zone, we represented the expression mean of dividing cells and cells from the quiescent center. See Section B.1 for details and Appendix A for the relationships between gene names and accession numbers.

**Figure 5 ijms-22-11043-f005:**
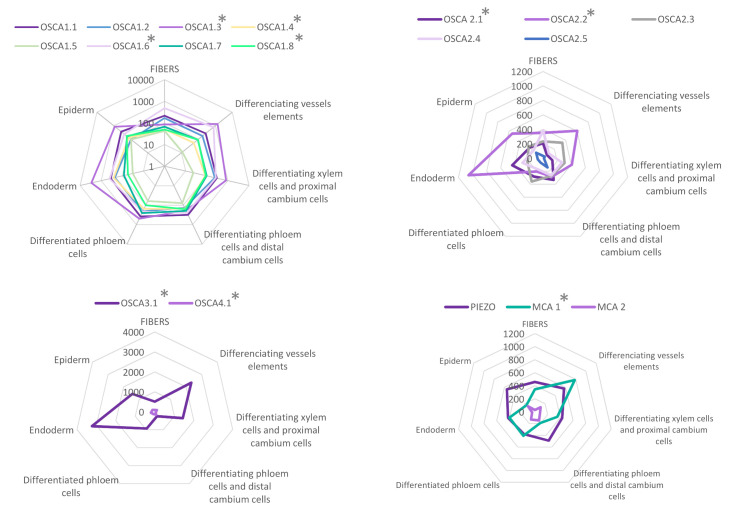
Expression patterns of *MCA*, *Piezo* and *OSCA* genes in different tissues of *Arabidopsis* mature inflorescence stem. Data were retrieved from the tissue-specific transcriptome study of the mature inflorescence stem of *Arabidopsis thaliana* [128] realized by analyzing mRNA from specifically labeled- nuclei covering a comprehensive set of distinct tissues (FANS/RNA-seq). Data are available at https://arabidopsis-stem.cos.uni-heidelberg.de (accessed on 20 June 2021). The values of expression levels (0 to 10,000) correspond to the mean of normalized read count values (by DESeq2). n=3 for each population. * indicates a *p*-value <0.01 in a Likelihood Ratio Test (LRT) results (by DESeq2) testing if genes have similar expression among the seven different tissues. See Section B.2 for details.

**Figure 6 ijms-22-11043-f006:**
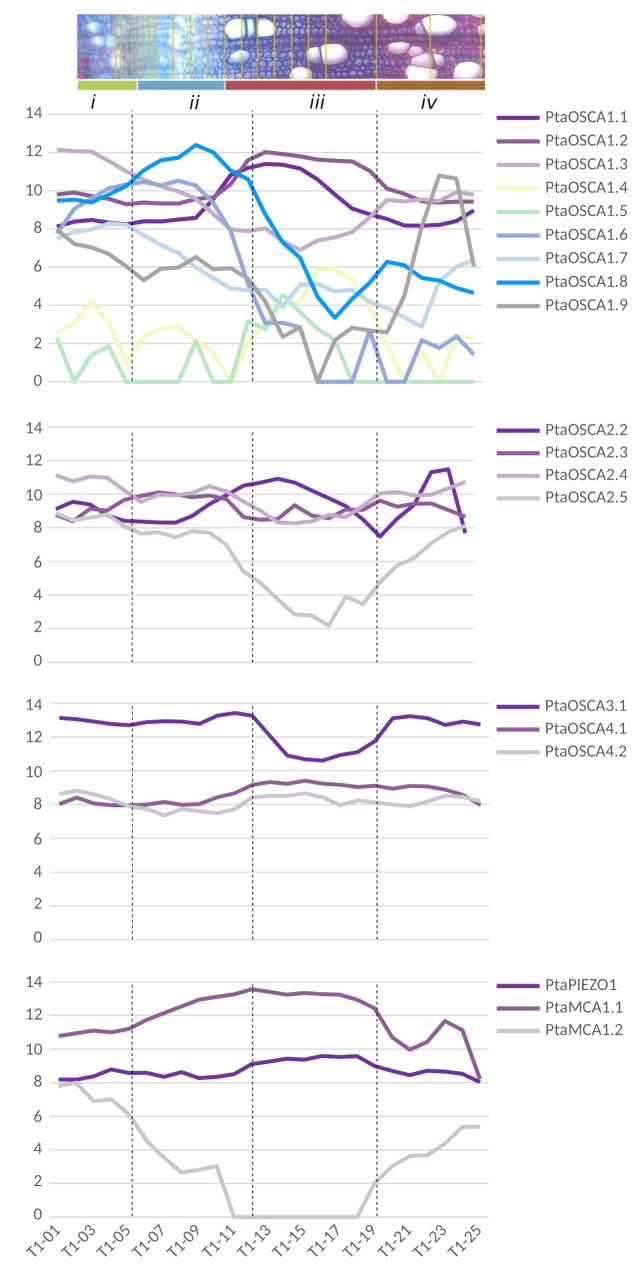
Expression patterns of *MCA*, *Piezo* and *OSCA* genes in the wood-forming tissues of a wild-growing aspen genotype (*Populus tremula*). Data were retrieved from Sundell et al. ([129]; Appendix A). Pooled longitudinal tangential cryosections were used to obtain a continuous sequence of samples extending from differentiated phloem to mature xylem, with values of expression levels ranging from 0 to 16 (see Section B.3 for details). To better illustrate the location of the different tangential cryosections pooled to produce an RNA-seq sample, we showed the transverse cross-section image from the sampled tree T1 from ([129], Figure 1). The pooled samples are visualized by overlaying them on the transverse cross-section and are grouped into four clusters (from *i* to *iv*). These clusters separate key reprogramming events represented by the dashed lines: (*i/ii*) occurs between phloem and xylem differentiation, in the middle of the dividing cambial cells; (*ii/iii*) marks the end of cell expansion and the onset of secondary cell wall formation; (*iii/iv*) marks the end of secondary cell wall deposition and the late maturation of xylem cells. The cambium zone is located in cryosections T1-04, T1-05 and T1-06.

## Data Availability

Expression patterns of *MCA*, *Piezo* and *OSCA* genes were retrieved from the BAR ePlantArabidospsis website (https://bar.utoronto.ca/eplant (accessed on 23 February 2021) for the *Arabidopsis* root, from https://arabidopsis-stem.cos.uni-heidelberg.de (accessed on 20 June 2021) for the *Arabidopsis* mature inflorescence stem, and from http://aspwood.popgenie.org (accessed on 20 June 2021) for wood-forming tissues in *Populus tremula*.

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
