# Peer review of "Between Stress and Response: Function and Localization of Mechanosensitive Ca2+ Channels in Herbaceous and Perennial Plants"

_ijms, 2021, doi:10.3390/ijms222011043_

Round 1

Reviewer 1 Report

Dear Editors,

Thank you so much for choosing me as a reviewer of the manuscript
ID ijms-1402284 entitled “Between stress and response: Function and localization of mechanosensitive Ca2+ channels in herbaceous and perennial plants” submitted  to International Journal of Molecular Sciences. I hope that my comments will help Authors to improve their manuscript.

Detailed remarks concerning manuscript:

Some methodology information should be provided. Please include the data concerning searched databases and the searched phrases based on which this review manuscript was written.

All Latin names of the species (see Oryza sativa, Plantago major, Arabidopsis thaliana and others) Arashould be italicized. It concerns not only the text of the manuscript, but also the titles of the manuscripts in the references. Please go through the whole text of the manuscript including references and do needed changes.

Reference list. Please do some editorial changes in the reference list. For instance once each word of the manuscript title is written with capital letter, but the other time only the first word of the manuscript title is written with capital letter. Please go through the whole reference list and do needed changes.

Reviewer 2 Report

In this review article the authors summarize the recent advances in the study of sensing and response of mechanical stress in plants. The text covers the brief description of mechanical stress and its effect on plant growth and development as well as the roles of mechanosensitive (MS) ion channels in mechanosensing. Reconstitution of RNA-seq data and visualization of tissue specific expression of MS channels of Arabidopsis and Populus is impressive and provides valuable insights into the function of the diverse channels. I also think all the cited literatures are appropriate and adequate.

However, I’d like the authors to check and modify several minor points as follows.

1) L174-175; I think “be” is missing in the phrase “could by mechanically coupled”

2) L271-272; “ED hand-like motif” should be changed to “EF hand-like motif”.

3) L573-575; The authors use “overexpressed” and “overexpression” to describe upregulation of genes. I think “overexpression” is usually used for artificial elevation of gene expression by transgenic approach. I would suggest using “upregulation” instead of “overexpression” in this context.

4) L666-667, Leek et al., 2018; This literature is not listed in References.

5) I think some information are missing for Ref 81.

6) There are several tags to be removed (such as <i>) for Ref 85, 102, and 136.

Reviewer 3 Report

The manuscript deals with the "Stress responses in model plants". In overall, manuscript has been written well. 

1. "Before the "1. Mechanical stresses and responses in plants", write a short INTRODUCTION. What is the lack of previous studies? problem statement? Gap of the knowledge?

2. Quality of Figure 6 should be improved.
